# Alternative Anti-Infective Treatments to Traditional Antibiotherapy against Staphylococcal Veterinary Pathogens

**DOI:** 10.3390/antibiotics9100702

**Published:** 2020-10-15

**Authors:** Álvaro Mourenza, José A. Gil, Luis M. Mateos, Michal Letek

**Affiliations:** 1Departamento de Biología Molecular, Área de Microbiología, Universidad de León, 24071 León, Spain; amouf@unileon.es (Á.M.); jagils@unileon.es (J.A.G.); 2Instituto de Biología Molecular, Genómica y Proteómica (INBIOMIC), Universidad de León, 24071 León, Spain; 3Instituto de Desarrollo Ganadero y Sanidad Animal (INDEGSAL), Universidad de León, 24071 León, Spain

**Keywords:** *Staphylococcus*, animals, bacteriocins, bacteriophages, host-directed therapies, probiotics, prebiotics

## Abstract

The genus *Staphylococcus* encompasses many species that may be pathogenic to both humans and farm animals. These bacteria have the potential to acquire multiple resistant traits to the antimicrobials currently used in the veterinary or medical settings. These pathogens may commonly cause zoonoses, and the infections they cause are becoming difficult to treat due to antimicrobial resistance. Therefore, the development of novel alternative treatments to traditional antibiotherapy has gained interest in recent years. Here, we reviewed the most promising therapeutic strategies developed to control staphylococcal infections in the veterinary field to overcome antibiotic resistance.

## 1. Introduction

Over the past few years, the one health concept has increased significantly in importance to understand the evolution of antimicrobial resistance. This concept has evolved from one *medicine* to one *health*, and it refers to the increasing number of pathogenic bacteria affecting both animals and humans [1]. Indeed, the number of epizootic diseases and zoonoses is increasing. Thus, animal and human health should be treated jointly to avoid the spread of antimicrobial resistance.

Members of the genus *Staphylococcus* are common causative agents of both human and animal infections [2] (Table 1). Moreover, staphylococcal infections are related to the development and spread of antimicrobial resistance. Indeed, Methicillin-Resistant *Staphylococcus aureus* (MRSA) is one of the most important multidrug-resistant pathogens in humans and animals [3]. Thus, the use of antibiotics against staphylococcal infections in animals can trigger the emergence of antimicrobial-resistant strains in humans [4]. Besides, the number of pan-susceptible staphylococci is decreasing substantially, and low-pathogenic *Staphylococcus* (coagulase-negative strains) may be the reservoirs of resistant genes [2,5]. Consequently, it is estimated that the number of resistant strains will increase substantially in the near future. Thus, the development of new anti-infective strategies to improve animal care is becoming urgent.

Staphylococcal infection in animals is caused by different species of *Staphylococcus* that may affect a variety of domestic and livestock animals. *Staphylococcus* spp. are part of the normal cutaneous and mucosal microbiota of mammals and birds [2]. Most animals are colonized by one or more *Staphylococcus* spp. during their lives and in different body parts [6,7]. There are numerous pathogenic species of *Staphylococcus*, classified as high, medium, or low-pathogenic bacteria. Their virulence potential depends on their ability to form biofilms, circumvent the host immune system, colonize different environments in the host, their toxin production levels, and their ability to survive internalization in host cells [8]. The development of new therapies against staphylococcal infections should be designed considering the diversity of opportunistic pathogens in the genus. The most pathogenic species of the genus is *S. aureus*, but *Staphylococcus epidermidis* has emerged as an important pathogen in humans and animals in recent years [8]. Moreover, other species have emerged as opportunistic pathogens such as *Staphylococcus saprophyticus*, *Staphylococcus haemolyticus*, *Staphylococcus lugdunensis*, *Staphylococcus hycius,* and *Staphylococcus pseudintermedius*, among others. In particular, *S. pseudintermedius* is a common cause of pyoderma in dogs and other animals [2,8,9]. In farm animals, coagulase-negative staphylococci such as *Staphylococcus chromogenes* are commonly causing subclinical mastitis in dairy cattle [10].

Any attempts to develop an opsonic antibody-based vaccine against staphylococci have failed up to now [11]. Fortunately, there are different alternatives not based on antimicrobials that could be used to control staphylococcal infections in animals. These alternative therapeutic strategies may reduce the number of multidrug-resistant strains in human and animal populations and their zoonotic transmission.

## 2. Treatments Based on Feed Supplements

Different treatments based on natural compounds are available to control staphylococcal mastitis, one of the leading causes of disease in dairy animals [13]. However, the use of antibiotics is still the most common treatment, which has led to the emergence of Livestock Associated MRSA (LA-MRSA) [14].

A promising solution to this problem is based on the use of antimicrobial peptides produced by generally recognized as safe (GRAS) bacteria [15,16,17]. Nisin is a bacteriocin produced by *Lactococcus lactis* with antimicrobial activity against several animal pathogens, including *S. aureus* [13,18] (Table 2). Moreover, bacteriocins can inhibit *S. aureus*’ biofilm formation, especially relevant to host colonization [18,19].

Several natural variants of nisin are produced by *Lactococcus lactis* and different species of *Streptococcus* [42], whose biomedical application has been extensively studied. Nisin A was the first discovered bacteriocin, but new nisin variants have been found [42]. Despite the fact that the antimicrobial effect of nisin variants has been demonstrated [13,18,19,20,21], very few scientific studies have focused on their possible use against subclinical mastitis [43]. The same applies to plantaricin NC8 αβ, a bacteriocin produced by *Lactobacillus plantarum* with great heat and pH stability [44].

Moreover, *Bacillus thuringiensis* is a very well-known producer of bacteriocins with antimicrobial activity against several *S. aureus* strains, morricin 269 being the most active bacteriocin against this pathogen. In addition, kurstacin 287, kenyacin 404, entomocin 420, and tolworthcin 524 also show a certain degree of activity against *S. aureus* [13,21].

Other very promising bacteriocins are colicins and pyocins, which are small proteins (~100 amino acids) used by bacteria for intraspecies competition [15]. Colicins are produced by *Escherichia coli* and pyocins by *Pseudomonas aeruginosa*.

Interestingly, antimicrobial peptides (AMPs, [45]) are naturally occurring peptides with antimicrobial activity that are produced by fungi, plants, amphibians, crustaceans, birds, and mammals, and could also be synthetically produced [46]. AMPs have shown activity against the most important ESKAPE pathogens (*Enterococcus faecium*, *Staphylococcus aureus*, *Klebsiella pneumoniae*, *Acinetobacter baumannii*, *Pseudomonas aeruginosa*, and *Enterobacter* species) [16,17,46,47]. Overall, AMPs could be an easy and low-cost therapy against staphylococcal infections in farm animals.

Similarly, cyclotides are plant-derived peptides with an antibacterial activity produced by a large variety of species [48,49,50]. Their antimicrobial activity against staphylococci was demonstrated 20 years ago [51]. Two different cyclotides (Table 2), cycloviolacin 2 and kalata B2, have been used to treat cellular infection caused by *S. aureus* in RAW 264.7 monocytes and mice, without producing any cytotoxicity against host cells [22].

In parallel, the use of probiotics and prebiotics in animal healthcare has increased during the last few years (Table 2). Probiotics are living organisms recognized as safe that could be used as part of the animal diet (see Section 3.2), whereas prebiotics are molecules that elicit the growth of natural gut microbiota. Both pro- and prebiotics are natural strategies implemented to fight multidrug-resistant bacteria in animal feeds [52,53].

Prebiotics could be used to elicit the growth of natural commensal bacteria that may exclude staphylococci colonization of animals (Table 2). However, the use of prebiotics in animal feed has led to a variable efficacy, and not always resulted in changes in the natural gut microbiota [35]. The most promising prebiotics for animal healthcare are nondigestible oligosaccharides (NDOs) such as galacto-oligosaccharides (GOS), fructo-oligosaccharides (FOS), cello-oligosaccharides (COS) or mannan-oligosaccharides (MOS), among others, which stimulate the growth of beneficial commensal bacteria in the animal intestine [23,35,54,55]. In particular, NDOs have been tested in pigs, ruminants, and poultry, where they elicit the growth of Bifidobacteria and Lactobacilli and increase the host immunological defenses [23]. Similarly, the plant-derived inulin or anthocyanins elicit the proliferation of probiotic bacteria, which potentially leads to the exclusion of *S. aureus* in the host [25]. In particular, inulin could be used as a feed additive in poultry to prevent *S. aureus*’ colonization [23,24,54,56]. However, an inulin-based feed may generate intestinal damage in the gilthead seabream [57]. Therefore, the use of prebiotic should be evaluated individually to exclude any adverse effects in the host.

Feed supplements could also be based on inorganic molecules with antimicrobial activity, such as zeolites (e.g., clinoptilolite) with antimicrobial activity against *S. aureus* [26]. Other compounds that could be an excellent solution to prevent staphylococcal infections and bacterial resistance are phenolic compounds such as resveratrol or dihydroquercetin [27,28,29] (Table 2). In particular, resveratrol is an efflux pump inhibitor that increases the efficacy of other antimicrobials [27]. However, resveratrol is an antioxidant compound [58], and therefore, its use in combination with oxidative stress-generating antimicrobials could decrease the efficacy of the anti-staphylococcal therapy [59].

## 3. Treatments Based on the Use of Other Microorganisms

### 3.1. Phage Therapy

Bacteriophages, also known informally as phages, are viruses that infect bacteria. Bacteriophage-based therapies are a century-old approach to control bacterial infections [60,61]. However, their clinical use was discarded after the discovery of antibiotics due to their high strain specificity, low stability, and the rapid development of bacterial resistance to specific bacteriophage-based therapies. Antimicrobial resistance has renewed the interest in developing treatments against multidrug-resistant bacterial infections based on bacteriophages [60,61]. Indeed, phages do have several advantages over traditional antimicrobial treatments. 

In particular, phages are highly specific, and therefore, bacteriophage-based therapies do not affect the normal flora nor eukaryotic cells. Besides, these are low dosage treatments, they rapidly proliferate inside the host bacteria, and they could be obtained from the natural environment, which makes them a perfect therapy against antimicrobial-resistant infections in animals [60,61]. Moreover, unlike antibiotics, bacteriophages can control putative reinfections, and they may mutate alongside their hosts to prevent the apparition of resistant bacteria [62]. 

However, the use of bacteriophages requires a detailed study of their interaction with targeted bacteria due to their strain specificity [53]. Therefore, monophage therapy must always be preceded by an in vitro assay to study the efficacy of the selected bacteriophages against the specific bacterial strain causing the disease [60]. Moreover, some bacteriophages may even introduce or activate virulence genes into the bacterial genome [53]. 

Despite these negative aspects, bacteriophages have already been used against staphylococci in animals, frequently in phage cocktails [61,63,64]. Interestingly, the bacteriophage cocktails formulated against the dog pathogen *Staphylococcus pseudintermedius* have shown activity against other pathogen species of *Staphylococcus*, including *S. epidermidis*, *S. haemolyticus*, and *S. saprophyticus* [65,66], suggesting that many bacteriophage-based therapies could be repurposed against different staphylococcal infections. In addition, the lytic bacteriophage phiSA012 and its endolysin Lys-phiSA012 (Table 2) have different veterinary *Staphylococcus* sp. targets [30], which may pave the way to develop pan-staphylococcal phage-based therapies.

### 3.2. Competitive Exclusion of Pathogens

Probiotics can inhibit colonization of the gut by pathogens by direct competition or by the production of toxic substances (i.e., bacteriocins) [67,68,69,70]. For instance, natural skin microbiota can inhibit the colonization of the host by staphylococci. Indeed, the external application of beneficial bacteria is essential for preventing and treating staphylococcal skin infections in atopic dermatitis’ patients [71]. Besides, probiotics reduce the concentration of methicillin-resistant *S. aureus* in farm animals [72]. Many strains of the genera *Lactobacillus* and *Lactococcus* can produce bacteriocins that are active against staphylococci [34,35,42,73] (Table 2). In addition, many of these lactic acid bacteria are resistant to antibiotics that could be used in combination with these probiotics and enhance their antimicrobial activity [34]. 

Moreover, the list of bacteria and fungi that could be used as animal probiotics is increasing. *Lactobacillus*, *Lactococcus,* and *Bifidobacterium* are the most commonly used, but different species of *Enterococcus*, *Leuconostoc, Pediococcus*, *Propionibacterium*, *Streptococcus*, *Bacillus*, *Saccharomyces*, *Kluyveromyces,* and *Aspergillus* have shown very promising probiotic assets [35,36]. Interestingly, it has been discovered that a *Bacillus* sp. probiotic strain produces a lipopeptide capable of inhibiting the quorum-sensing signaling system of *S. aureus*, which abolishes the pathogen’s colonization of the human gastrointestinal tract [74]. Similar strategies based on quorum-sensing blockers may be used in animal health in the near future. 

However, the precise mechanism of action of many of these probiotics is still not well understood. Also, the significant dose of probiotics required to cause an effect on the host, combined with their poor mucosal colonization, have hampered the development of animal feed supplements based on this strategy [67,69,75]. Therefore, more research must be done before extending the use of probiotics in animal feed. Importantly, gut microbiome changes must be thoroughly analyzed after antibiotic treatments because some changes in their composition could alter the beneficial effects of probiotics [67,68].

An interesting alternative to probiotics has recently emerged in the form of predatory bacteria against several important human pathogens [76] (Table 2). Many of the *Bdellovibrio* and *Micavibrio* species described are common Gram-negative predatory bacteria. However, they can also disrupt biofilms created by *S. aureus*, which facilitates the antimicrobial response of macrophages against this pathogen and liberates free amino acids in the process [77,78]. Other predatory bacteria, such as those belonging to the genus *Herpetosiphon,* produce secondary metabolites with antimicrobial effects and show prey activity against *S. aureus*, *S. epidermidis,* and *S. saprophyticus* [31]. Finally, some predatory bacteria, such as *Myxococcus xanthus*, may secrete outer membrane vesicles (OMV) that carry antimicrobial compounds and proteins active against *S. aureus* and *S. epidermidis* [31,32,33].

## 4. Other Alternative Anti-Infectives Against Staphylococci

### 4.1. Host-Directed Therapies against Staphylococcus spp. Infections

Host-directed therapies (HDT) against *Staphylococcus* species are still poorly studied in farm animals. However, the host response during staphylococcal infection is well known in some animal models (in particular rabbits and cattle), and this knowledge could pave the way to develop novel host-targeted therapies [10,79].

For example, dairy cows suffer a reduction in immune function during lactation, with a subsequent predisposition to infectious disease [37,80]. This immunosuppression could be reversed to avoid staphylococcal infections with pegbovigrastim, a recombinant DNA-derived bovine granulocyte colony-stimulating factor (G-CSF) analog (Table 2). G-CSF-based treatments increase the number of neutrophils with bactericidal activity [37,80,81,82].

On the other hand, diet and nutritional supplements based on garlic have many advantages for animal health, particularly in poultry [83]. Some secondary metabolites present in garlic enhance the humoral immunity in chickens and elicit a direct antimicrobial effect against *S. aureus* by generating oxidative stress [38,58] (i.e., allicin; Table 2) [84,85]. In addition, different probiotics elicit a potent cellular and humoral immune response against staphylococci by increasing the production of IgG, IgM, and IgA [86,87].

### 4.2. Immunotherapies

Immunotherapies are well-established treatments in cancer therapy that are now considered a very promising strategy to control infections caused by multidrug-resistant bacteria [88]. In particular, immunotherapies are based on the use of antibodies that may interfere with the pathogenicity of the bacteria [89]. 

For example, monoclonal antibodies targeting bacterial membrane proteins called adhesins are efficient treatments against *S. aureus* infections by inhibiting the pathogen’s adhesion to host cells and tissues and its subsequent colonization [12]. Another important target of monoclonal antibodies could be specific membrane proteins of eukaryotic cells that act as receptors of toxins produced by the pathogen during infection. These toxins may generate necrosis in the infected tissues to release the intracellular contents of host cells and increase the availability of nutrients and growth factors in the interstitial fluid [90].

However, the current high cost of immunotherapies is still a critical factor that hampers their application in farm animals, despite its up-and-coming applications. Consequently, immunotherapy has only been applied in small animals such as cats or dogs [91,92], and it is unclear if this strategy could be eventually applied in animal production at a reasonable cost.

### 4.3. Small-Interference RNAs (siRNAs)

Another strategy to fight against multidrug-resistant staphylococci infections in animals could be based on the use of siRNAs. Similarly to some immunotherapies, siRNAs could be used to silence the expression of specific virulence factors in the pathogen by targeting their messenger RNAs [87,88]. Importantly, siRNAs can be synthetically designed to match the sequence of any gene, and therefore, this is a very versatile strategy to combat multidrug-resistant bacteria. siRNAs-based therapies have been proposed as a putative solution against MRSA in humans by targeting the coagulase expressed by the pathogen [39]. However, other targets will have to be validated to control infections caused by coagulase-negative staphylococci. Besides, little is known about the applicability of this strategy to control staphylococcal infections in animals. Synthetic siRNAs have been designed to silence targets of viruses that cause diseases in cats [93]. Interestingly, a combination of different siRNAs may show additive or synergistic interactions [93].

However, the efficient delivery of siRNAs to their targeting cell is still one of the significant limitations of this strategy because siRNAs could be degraded by extracellular nucleases present in host tissues [94,95]. Therefore, new strategies have been developed to improve siRNAs’ delivery, including different conjugation methods with immunoproteins, peptide ligands, or aptamers [95,96,97]. Besides, siRNAs could also be encapsulated in gold or silver-based nanoparticles, liposomes, or cyclodextrin complexes that may protect their cargo against nuclease attacks [95,96,97].

### 4.4. Nanoparticles

Metal nanoparticles have recently emerged as another alternative strategy for treating bacterial infections. In particular, zinc oxide nanoparticles are considered a safe and low-cost therapy against many human infections [40,98] (Table 2). Interestingly, nanoparticles have shown antimicrobial activity against intracellular pathogens because they may stimulate the biosynthesis of Reactive Oxygen Species (ROS) within subcellular compartments of the infected host cell, causing oxidative stress and damage to bacterial membranes [41,58]. Zinc oxide nanoparticles have broad-spectrum antimicrobial activity, and they may be useful in treating MRSA infections [40,41]. Additionally, the combination of pancreatin and zinc oxide nanoparticles showed synergy against MRSA by simultaneously affecting multiple virulence factors, biofilm formation, and pathogen growth [40]. Consequently, the use of zinc oxide nanoparticles against infectious diseases is now widely accepted in animal production [99,100].

However, the extensive use of heavy metal nanoparticles could also risk the co-selection of heavy metal resistance genes and antimicrobial resistance genes [101,102], which may aggravate the problem of antimicrobial resistance in the long term. In addition, the release of zinc to the environment, a highly contaminating heavy metal, is considered a significant limitation of this strategy.

## 5. Conclusions

The one-health concept is of major importance for the rational design of strategies centered on controlling the spread of antimicrobial resistance. It is becoming clear that the abuse of antimicrobials in animal production results in an increase of multidrug-resistant infections in humans. Any alternative to using antimicrobials in the control of animal infections will have a significant impact on human health. Here, we have summarized the most promising anti-infective therapeutic strategies to treat staphylococcal infections that could be used in animal health as an alternative to antibiotics. Many of the treatments covered in this article are currently considered effective, safe, and low-cost treatments, and may reduce the use of antimicrobials in animal production in the long term. In particular, the use of phage cocktails or probiotics may be a natural and inexpensive solution to the problem of antimicrobial resistance in animal health. However, further research is needed to implement these anti-infective strategies in animal production at the industrial scale.

## Figures and Tables

**Table 1 antibiotics-09-00702-t001:** List of pathogenic staphylococcal species and the diseases they may cause in humans and animals.

Species	Host	Disease	References
*S. aureus*	Humans	Bacteremia; skin abscesses; severe chronic infections	[3,12]
	Dogs and cats		[3]
	Horses		[3]
	Cattle	Mastitis	[3]
	Poultry	Skeletal infections	[3]
*S. chromogenes*	Cattle	Subclinical mastitis	[10]
*S. epidermidis*	Humans	Septicemia	[8]
	Domestic animals	Bacteremia	[8]
*S. haemolyticus*	Humans	Hemolysis	[2]
	Cats and other small animals	Hemolysis	[2]
*S. hycius*	Pigs	Epidermitis	[2]
*S. pseudintermedius*	Dogs	Pyoderma	[2,8,9]
*S. lugdunensis*	Humans	Acute skin and soft tissue infections; bacteremia	[2,8]
	Domestic animals	Acute skin and soft tissue infections; bacteremia	[2,8]
*S. saprophyticus*	Humans	Urinary infections	[7,8]

**Table 2 antibiotics-09-00702-t002:** List of the most promising alternative anti-infective treatments to traditional anti-biotherapy against staphylococci.

Treatments	Examples	Tested Species	Model System Used to Test the Effect	Outcome Measure(s)	References
Antimicrobial peptides (AMPs)	Bacteriocins	*S. aureus*	In vitro	Curative	[13,18,19,20,21]
	Cyclotides	*S. aureus*	Animals		[22]
	Other AMPs	*S. epidermidis*	Humans		[16]
Prebiotics	Non-digestible oligosaccharides	*-*	Animals	Preventative	[23]
	Inulin	*S. aureus*	Animals		[24]
	Anthocyanins	*S. aureus*	In vitro		[25]
Zeolites	Clinoptilolite	*S. aureus*	Animals	Curative	[26]
Polyphenols	Resveratrol	*S. aureus*	Animals	Curative	[27,28,29]
	Dihydroquercetin	*S. aureus*	Animals		[27]
Bacteriophages	*phiSA012*	*Staphylococcus* spp.	In vitro	Curative	[30]
Predatory bacteria	*Herpetosiphon* sp.	*S. aureus* *S. epidermidis* *S. sparophyticus*	In vitro In vitro In vitro	Curative	[31]
	*Myxococcus xanthus*	*S. aureus* *S. epidermidis*	In vitro In vitro		[31,32,33]
Probiotics	*Lactobacillus* sp.	*S. aureus*	Animals	Preventative	[34]
	*Lactococcus* sp.	-	Animals		[35,36]
	*Bifidobacterium* sp.	-	Animals		[35,36]
	*Enterococcus* sp.	*S. aureus*	Animals		[36]
Host-directed therapies	Granulocyte colony-stimulating factor	*S. chromogenes*	Animals	Preventative	[37]
Secondary metabolites derived from plants	Garlic	*S. aureus*	In vitro	Preventative	[38]
Immunotherapies	Monoclonal antibodies	*S. aureus*	Humans	Curative	[12]
Transcriptional control	siRNAs	*S. aureus*	Animals	Curative	[39]
Nanoparticles	Zinc oxide nanoparticles	*S. aureus*	Ex-vivo	Curative	[40,41]

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
