# Peer review of "Alternative Anti-Infective Treatments to Traditional Antibiotherapy against Staphylococcal Veterinary Pathogens"

_antibiotics, 2020, doi:10.3390/antibiotics9100702_

Round 1

Reviewer 1 Report

Should be published in Antibiotics after minor revision

In their manuscript entitled “Alternative anti-infective treatments to traditional antibiotherapy against staphylococcal veterinary pathogens”, Álvaro Mourenza et al. focus on Staphylococcus spp. infections in the veterinary field, to review promising therapeutic strategies developed to overcome antibiotic resistance. As stated by Authors, this is an important topic to address, for both animal and human health (One health concept).

This review compiles more than one hundred references to relevant papers, most being published in the last few years. There is no inappropriate self-citation since among the references listed, only one (number 46) is from the same authors of the current paper. In my opinion, this paper is quite well-written and pleasant to read. After a brief update on the current situation of Staphylococcus spp. infections in the veterinary field, Authors go over potential alternative treatments to antibiotics. Although already interesting, I think this manuscript could be improved in different ways. I wonder whether other new strategies for the treatment of Staphylococcus infections could be implemented, in particular therapies specifically targeting regulatory networks (quorum-sensing). Furthermore, preventive and curative treatments could be better distinguished. The quality of Figures could also be increased and some paragraphs better written.

For all these reasons, I would recommend publication of this manuscript in Antibiotics after minor revision, taking into account the comments done above and further detailed below.

Main comments

- In Introduction, the first paragraph from line 45 to line 60 should be moved and merged with the second paragraph on page 1.

- In paragraph “2. Treatments based on feed supplements”, the discussion about antimicrobial peptides could be better conducted. Possible confusions are reflected in Figure 1, in which there is a lack of rigor in the delineation done. “Bacteria peptides” includes Cyclotides whereas these are isolated from plants, as Authors state on line 107. Another wording for “Organic molecules” should be used, since it could refer to other compounds than those considered.

- In Figures 1 and 3, the drawings provided for some compounds look useless, providing no relevant information for this paper. Moreover, there is no homogeneity (different types of representations are used, which are not always readable) and in some cases, there is no illustration at all. Thus, I would suggest removing these drawings and instead systematically add in an additional column the name of one example of compound for each type of family/strategy with reference(s) to studies in which this compound has been evaluated.

- Along the same line, Figure 2 could be noticeably improved.

- In Conclusions, though many of the alternatives treatments reviewed in this manuscript could indeed be applied to veterinary infections by different micro-organisms, authors should specify again the main subject of this review i.e. staphylococcal infections.

Minor comments

- On line 114, insert (see paragraph 3.2) after “the animal diet”.

- On line 200, [67][68] should be changed to [67,68].

- In Figure 3, correct “Feed suplement” (typo error).

Author Response

In their manuscript entitled “Alternative anti-infective treatments to traditional antibiotherapy against staphylococcal veterinary pathogens”, Álvaro Mourenza et al. focus on Staphylococcusspp. infections in the veterinary field, to review promising therapeutic strategies developed to overcome antibiotic resistance. As stated by Authors, this is an important topic to address, for both animal and human health (One health concept).

This review compiles more than one hundred references to relevant papers, most being published in the last few years. There is no inappropriate self-citation since among the references listed, only one (number 46) is from the same authors of the current paper. In my opinion, this paper is quite well-written and pleasant to read. After a brief update on the current situation of Staphylococcus spp. infections in the veterinary field, Authors go over potential alternative treatments to antibiotics. Although already interesting, I think this manuscript could be improved in different ways.

Many thanks for your kind comments and insightful suggestions.

I wonder whether other new strategies for the treatment of Staphylococcus infections could be implemented, in particular therapies specifically targeting regulatory networks (quorum-sensing).

Thanks for this suggestion; we have added information on quorum-sensing blockers in current lines 178-181.

Furthermore, preventive and curative treatments could be better distinguished.

We agree with the reviewer; we have added a column in Table 2 with this information.

The quality of Figures could also be increased and some paragraphs better written.

Following your suggestions in the main comments, we have replaced all three figures with Table 2, including much more valuable information.

For all these reasons, I would recommend publication of this manuscript in Antibiotics after minor revision, taking into account the comments done above and further detailed below.

Main comments

- In Introduction, the first paragraph from line 45 to line 60 should be moved and merged with the second paragraph on page 1.

This part of the introduction was modified, as suggested by the reviewer. The introduction is now more readable, thanks.

- In paragraph “2. Treatments based on feed supplements”, the discussion about antimicrobial peptides could be better conducted. Possible confusions are reflected in Figure 1, in which there is a lack of rigor in the delineation done. “Bacteria peptides” includes Cyclotides whereas these are isolated from plants, as Authors state on line 107. Another wording for “Organic molecules” should be used, since it could refer to other compounds than those considered.

As mentioned before, we have replaced all figures with Table 2, and there is a single category for antimicrobial peptides that includes bacteriocins, cyclotides, etc., to avoid confusion. The wording “Organic molecules” has been removed. We are referring now directly to polyphenols in Table 2.

- In Figures 1 and 3, the drawings provided for some compounds look useless, providing no relevant information for this paper. Moreover, there is no homogeneity (different types of representations are used, which are not always readable) and in some cases, there is no illustration at all. Thus, I would suggest removing these drawings and instead systematically add in an additional column the name of one example of compound for each type of family/strategy with reference(s) to studies in which this compound has been evaluated. Along the same line, Figure 2 could be noticeably improved.

All figures have been replaced with Table 2, as previously mentioned. We have included here one example of a compound for each type of strategy and references, following your suggestions.

- In Conclusions, though many of the alternatives treatments reviewed in this manuscript could indeed be applied to veterinary infections by different micro-organisms, authors should specify again the main subject of this review i.e. staphylococcal infections.

Thanks, staphylococcal infections are now mentioned in the conclusions (current line 281).

Minor comments

- On line 114, insert (see paragraph 3.2) after “the animal diet”.

- On line 200, [67][68] should be changed to [67,68].

- In Figure 3, correct “Feed suplement” (typo error).

Thanks, all of these comments have been addressed in the new version of the manuscript.

Reviewer 2 Report

This short review describes the alternative therapeutic potentials of Staphylococcal infections. It it summarized well but few things are not coherent:

  1. The title states 'veterinary pathogens' but such specificity is not noticed in the main article.
  2. The authors should consider adding a table showing different staphylococcal pathogens and the diseases they cause in different animals. 
  3. The 'One Health' concept brought in the introduction and then only at the conclusion do not go along with remaining parts of the article. Better reformatting of the manuscript is needed citing human and animal health perspectives, if need of 'One Health' approach is to be highlighted. 

Author Response

Many thanks for your revision of this manuscript. In response to points 1 and 2, we have included Table 1 in the new version of the review to list staphylococcal pathogens and the diseases they cause in animals and humans. With regards to point 3, many of these pathogens are now causing infections in multiple hosts, including human and animal populations, at the same time. Therefore, it is not easy to differentiate between animal and human pathogens when staphylococci are considered.

On the other hand, the effectiveness of many of the treatments reviewed here has only been tested in vitro. This information is now included in Table 2 to clarify the different infection models used to demonstrate the antimicrobial activity of the anti-infective strategies included in this revision. Nevertheless, it is becoming clear that many of these treatments should be applied in human and veterinary medicine, particularly in the case of preventative treatments. This is because the line differentiating human and animal health perspectives is becoming very blurred for this particular group of pathogens.

Round 2

Reviewer 2 Report

Thanks for the revision. 

Author Response

Many thanks for your comments.